# Human-Machine Interaction: Adapted Safety Assistance in Mentality Using Hidden Markov Chain and Petri Net

**Chao-Nan Chen [1], Tung-Kuan Liu [2],\* and Yenming J. Chen [3],\***

[1] Ph. D. Program in Management and in Engineering Science and Technology, College of Engineering, National Kaohsiung University of Science and Technology, Kaohsiung 824, Taiwan; mj.ceo@msa.hinet.net

[2] Department of Mechanical and Automation Engineering, National Kaohsiung University of Science and Technology, Kaohsiung 824, Taiwan

[3] Department of Logistics Management, National Kaohsiung University of Science and Technology, Kaohsiung 824, Taiwan

\* Correspondence: tkliuaiia@gmail.com (T.-K.L.); yjjchen@nkust.edu.tw (Y.J.C.); Tel.: +886-07-6011000 (Y.J.C.)



**Featured Application: This work can be applied to the situation of handling workers' stress in operating automatic machines.**

**Abstract:** This study proposes a cognition-adaptive approach for the administrative control of human-machine safety interaction through Internet of Things (IoT) data. As part of Industry 4.0, a human operator possesses various characteristics, but cannot be consistently understood as well as a machine. Thus, human-machine interaction plays an important role. This study focuses on incumbent challenges on the basis of estimated mental states. Given the operation logs from data recording hardware, a Hidden Markov model on top of a human cognitive model was trained to capture a production line worker's sequential faults. Our study found that retaining workers' attention is insufficient and tracking the state of perception is key to accomplishing production tasks. A safe workflow policy requires attention and perception. Accordingly, our proposed Petri Net enhances operation safety and improves production efficiency.

**Keywords:** Information processing model; hidden Markov chain; petri net; human–machine interaction

---

## 1. Introduction

Given the prominence of Industry 4.0, tasks and demands for humans in factories have changed [1]. As a critical entity in cyber–physical systems, human operators experience an increased complexity of jobs, such as specification and verification of production [2]. As an advocated focus of automation, human involvement has regained its importance, although the characteristics of making errors constantly persist. For factories categorized as safety-critical, such as those involving heavy machinery or hazardous substances, human intervention is deemed necessary to safeguard all occurrences of incidents and failures of processes and products. Hence, administration applies regulations to force workers to follow certain rules to prevent work hazards in the old era. However, humans are not machines and humans' internal cognition status frequently remains unknown. A single fit-all standard may not be practical and cost-effective because workers have different speed–safety tradeoffs in terms of capability and stress, such as fatigue. Therefore, aligning speedy workers (e.g., a robot) with slow ones (e.g., a human) is inappropriate. To assure the correct functioning of this critical component, an

assistance tool is essential for the successful implementation of Industry 4.0. In the post-Industry 4.0 era, we should address this challenge with an effective mechanism.

Work safety has become a major issue in man–machine coordination. However, the existing government and company safety rules cannot effectively prevent the occurrence of accidents. The reason is due to a wide range of dynamics, such as frequent product changes, staff turnover, and urgent delivery. Because heterogeneous operators with professional skills and trained mentality can enhance the flexibility, quality, and effectiveness of production, enterprises frequently use various staff training, on-site management, and auditing systems. In this case, people with different languages, cultures, and educational backgrounds in the same job site work together, thereby increasing the cost of education and difficulty of management.

Although the issues of human–machine interaction are considered important in practice and in academia, a scientific investigation of safety control rules remains limited. To fill in this research gap, this study aims to develop an effective set of safety rules based on data analytics and proposes an adaptive approach for administrative control of human–machine safety interaction through IoT (Internet of Things) data.

We facilitate a Hidden Markov model (HMM) on top of a human cognitive model to capture the sequential faults of a production line worker, who suffers from work stress. Data are collected from the recording of shaping operations of hydraulic machines in precision casting. The dynamic feature of the finite state machine is highlighted for heterogeneous human behavior. During the training stage, we estimate the state transition matrix on the basis of machine signals. During the operation stage, we assume the state of workers' attention solely based on molding sequencing. Management can enact a contingent safety rule, which may force a particular worker to rest when she or he is unable to meet the standard operation requirement.

## 2. Literature Review

Three streams of literature relate to this research. Many studies have proposed methods to improve work safety using Petri Nets (PNs) and Hidden Markov chains. Despite the sophisticated development of human cognitive theories, the integration of the human–machine perspective to the mentality is limited.

For the existing literature on human–machine failure analysis, numerous studies have concentrated on preventive mechanism in response to job actions. Dhillon and Yang [3] focused on the human–machine system and proposed a method of calculating the availability for situations with critical and non-critical human errors. Ram et al. [4] investigated a standby system with human operators through stochastic analysis. Longo et al. [5] promoted the safety assurance responsibilities of human-machine interaction to reinforce the capabilities and competencies of Industry 4.0 operators. Although intelligent mechanical safety has become an international trend, safety analysis through workers' mental states has been overlooked.

From the training viewpoint, Beck and Clark [6] studied training psychology to improve quality, safety, and efficiency. Staff members can meet job requirements safely after undergoing a comprehensive and continuous training. However, the cost of training materials, teachers, and time is increased inevitably. In particular, simultaneously coping with only a few orders is also difficult because of the memory and habits of the staff. From a management viewpoint, Gorecky et al. [2] found that management should meet the requirements of safety, efficiency, and quality. However, the additional management duties of inspection, auditing, and counseling vary from one person to another, while their effectiveness should be further investigated. A decision without going through a suitable perception state may cause dangerous consequences [7–9]. Risk management issues should be substantially addressed in market competition and production lines [10,11]. Safety issues in manufacturing industries are strictly regulated by the law. All companies should comply with the requirements of international safety norms, particularly safety supervision and management of mechanical products. Stringent and

standard-setting organizations, which are based on the current technological development, proposed an updated security design.

The concept of reliability has been integrated with intrinsic mechanical safety. In the future, mechanical safety control systems will be based on reliability-based and functional safety levels. This topic has attracted heated international discussions for mechanical safety standards [5]. Quality and environmental protection should be combined with safety and health and indispensably become a complete standard system. Statistical data on labor disability accidents over the years indicated that workers have suffered from "crippled, crushed and wound" handicapped cases, which are generally caused by power punching machines [3]. Safety protection, safety of equipment, control circuit, and safety warning information can be used to protect workers' safety owing to the various dangers of punching machinery.

Fault tree analysis (FTA) may be the most accepted technique among all safety or failure analysis methods [11]. The widespread use of FTA is due to its simplicity, although such simplicity also has several drawbacks. The most prominent drawback is that FTA's limited modeling representation can only handle logical relations [12,13].

To complement the static aspect of FTA, PN is a powerful tool and can effectively describe the dynamic behavior of a system. PN is a dynamic modeling tool and can capture unplanned failures and their sequence, thereby predicting the quality and reliability impact in a dynamic manner [14]. The modeling power of PNs is versatile. For example, Sachdeva et al. [15] and Caterino et al. [16] utilized PNs to analyze the reliability of a pulping system and a guillotine machine for retaining safe activities.

Given the academic and practical importance of safety management under working stress, this study concentrates on fault analysis on the basis of workers' cognitive states. However, estimating decision behavioral characteristics in a risk choice process has long been considered challenging. The perceptions of risk of decision makers are not straightforward because of the cognitive limitations of human perception. Quo et al. [17] considered human irrationality and compensated for human operators' decision bias on the basis of prospect theory. Chen et al. [18] offered a truth-telling choice to uncover the mind of human agents for accurate risk management.

Furthermore, Beck and Clark [6] proposed a basic model for information processing and used such a model to address the anxiety of human operators. Piechulla et al. [19] developed an adaptive human–machine interface to properly alleviate a driver's mental workload through an information processing model collected from traffic situations and driving dynamics. Similar to this study, a threshold alert will be devised to ensure the safety of normal operations. Brookhuis and Waard [20] also maximized the physiological measures of drivers to estimate their mental workload using a driving simulator. Galy et al. [21] explored the relationship between mental workload and cognitive load to obtain further understanding of psychological dynamics. Similarly, Robinson et al. [22] emphasized that the gaps between perceived risks (attitudes) and measurable risks (responses) should be explained on the basis of the characteristics of risk, psychological dynamics, or socio-demographic factors. In the next section, we will describe how we use a cognition-adaptive approach for administrative control of human-machine safety interaction using IoT data.

## 3. Method

### 3.1. Problem Description

The problem deals with human–machine coordination through an operator's mental state in a manufacturing process. Given a combination of multinational workers who are operating dangerous machines, the design, education, training, and management of the operation becomes complicated. When operating a shaping machine, a worker will press the machine multiple times to form the working objects to the desired shape following a certain accuracy. The number of presses in a focus state should be less than that in a fatigue state. The currently implemented safety protection is limited because of the need for sophisticated operation and accuracy. Workers are instructed to follow a standard

operating procedure, as shown in Figure 1. A conventional raster grating switch or two-hand button switch method cannot completely meet the safety requirements. For example, operating sculpture, pottery, and other manual work require the manual adjustment of the workpiece angle and position. In some situations, the operator's hand could accidentally enter the work area and cause an injury. Therefore, we start our first trial to map a person's mind to external measurement without physically binding an equipment in order to achieve unobtrusive personnel safety.

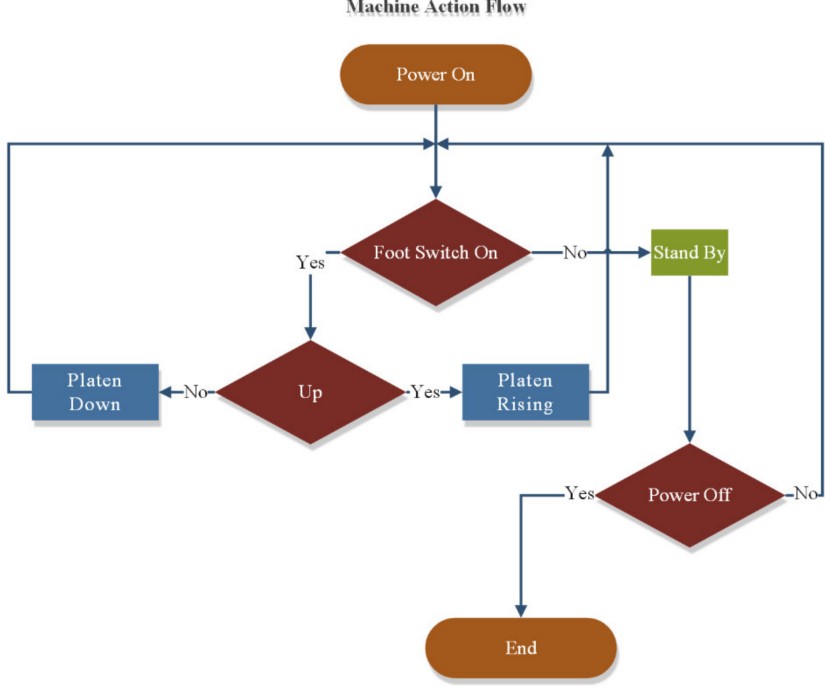

**Figure 1.** Standard operating procedure for a shaping machine.

This study collects a large amount of machine data through installed Internet of things (IoT) devices. The time epochs in moving the pressing mold down and up are recorded into a stream database. Figure 2 shows the distribution pattern of the pressing duration for a machine within a particular day. The horizontal and vertical axes represent the number of presses and time duration to the next press, respectively. The duration looks deviated on the first half day and uniform at the second half day, although the majority look scattered. Accordingly, judging the mental state of a particular worker simply by inspecting the changes in stimulus–response is difficult. Hence, a deliberate approach should be developed to monitor the mental state of workers.

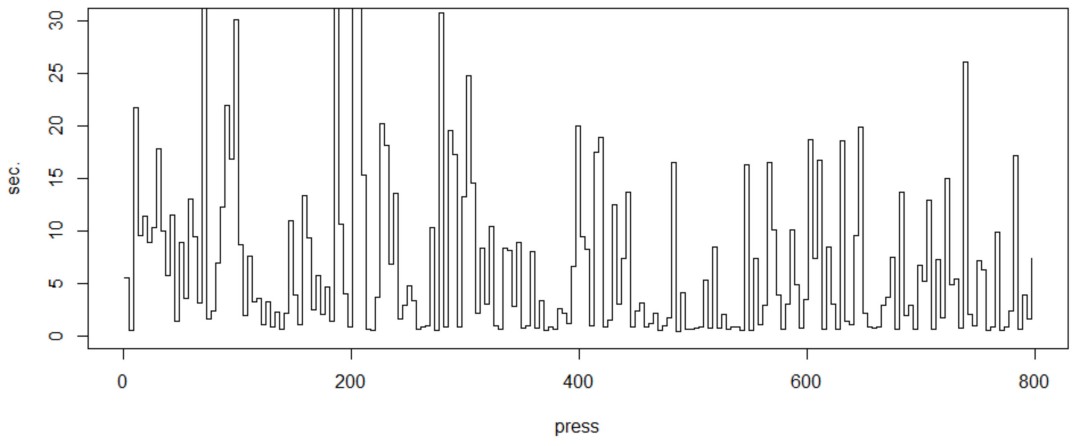

**Figure 2.** Distribution pattern of pressing duration within a given period.

### 3.2. HMM of Cognitive States

We facilitate the Hidden Markov Model (HMM) on top of a human cognitive model to capture the cognitive states of a production line worker who suffers from working stress. The HMM is deemed a double stochastic model, which converts the serial correlation of observable data into a probability transition of latent states [23]. The states of a system are hidden to the observer. However, by learning through advanced algorithms, determining which state the system is most likely when an output is observed is possible. The model is maximized when the states are indirectly visible but should be estimated through visible process outputs. Each state has a known probability distribution over the possible outputs. That is, the sequences of outputs generated by an HMM state transition reveal information on incumbent states.

An HMM can be formally characterized by five elements. $N$ is the number of states in the model. The state values in the state variable $q_t$ are denoted as $S = \{S_1, S_2, \cdots, S_N\}$. $M$ is the number of distinct observation symbols for the outputs of an HMM. The symbols are denoted as $V = \{V_1, V_2, \cdots, V_N\}$. A transition probability matrix $(A = \{a_{ij}\})$ indicates the likelihood of transitioning from one state to another, where $a_{ij} = \text{Prob}(q_{t+1} = S_j | q_t = S_i), 1 \le i, j \le N$ represents the probability value of state $q_t$ of going to state $S_j$ at time $t + 1$, given that at time $t$, the state is $S_i$. An output probability matrix $(B = \{b_j(k)\})$ indicates the likelihood for a certain measure value to come from a given state, where $b_j(k) = \text{Prob}(V_k(t) | q_t = S_j), 1 \le j \le N, 1 \le k \le M$ presents the probability of the $k$ th observation symbol because the state is state $S_j$ at time $t$. Lastly, an initial state distribution $\pi_1 = \{\pi_{i1}\}$ indicates the likelihood for a new input sequence to start in a given state at the initial time $t = 1$, where $\pi_{i1} = \text{Prob}(q_1 = S_i), 1 \le i \le N$. The sum of all probability distributions must be equal to 1, while the sum of all elements in a row in the $A$ matrix must also be equal to 1.

The dynamic feature of the finite state machine is highlighted for heterogeneous human behavior. This study estimates an operator's mental state through an HMM, which is a Markov process with unobserved (hidden) states. During the training stage, we estimate the state transition matrix according to the measured molding sequencing, personal traits, and annotative mind states. During the operation stage, we estimate the state of workers' attention solely based on measured molding sequencing. We will formulate a contingent safety rule, which forces a suspected worker to rest when she or he is unable to surpass the safety standard under continuous operation.

### 3.3. Dynamic State Estimation from IoT Records

Molding sequencing data are collected from an actual production line with a shaping operation of a hydraulic machine in precision casting. Recording hardware logs the molding sequences and administration software records worker identification, gender, and seniority. The data consist of a long sequence of time epoch pairs $(T_{down}, T_{up})$, which represent the timing and durations in pressing operations.

We first converted the absolute time epoch of pressing to a time duration between two presses. We use this sequence of durations to estimate the transition matrix of HMM. To justify the mind state shifts, the entire course of records in the distribution pattern of Figure 2 was divided to five time-blocks in order to investigate workers' fatigue or stress in a time interval. For the convenience of applying Viterbi algorithm, we discretized the pressing durations into five stages such that each stage carries almost equal weight in histogram.

The information processing model in Figure 3 indicates the number of cognitive states suggested by previous research [6]. To further explain the Markov chain fitting results, we simply assume a minimal number of states to manifest the changes in state distributions. For each time-block of sequences, a state transition matrix with three states is estimated through the observed data. That is, for each time block, an HMM operates on three states with five emissions. The state transition and emission are depicted in Figure 4.

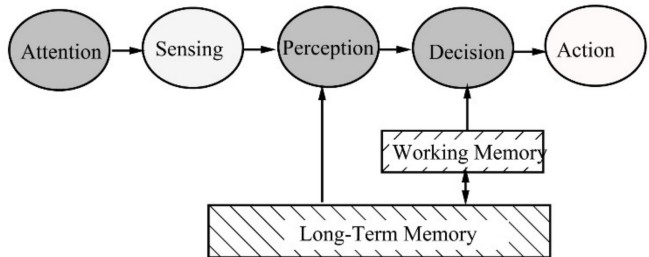

**Figure 3.** Information processing model.

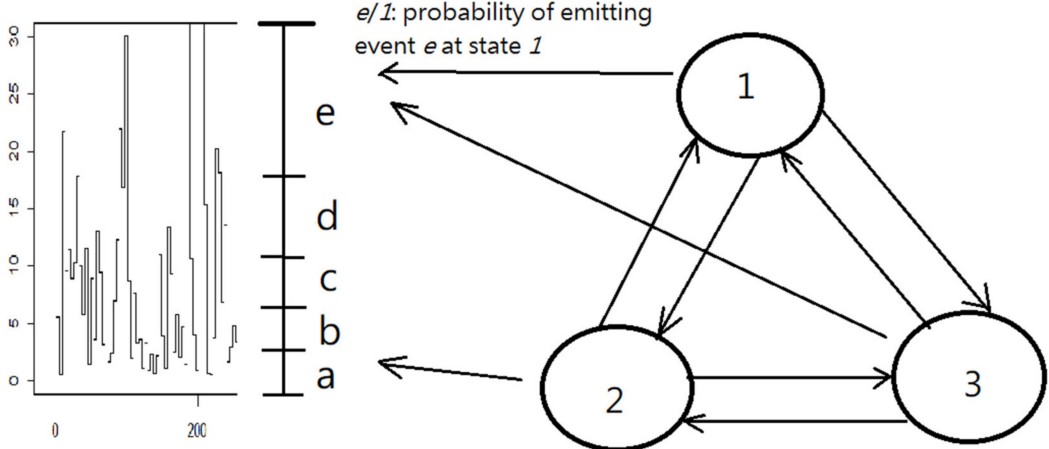

**Figure 4.** Hidden Markov Chain and Emission to Observations.

Up to this point, we obtained a probability transition matrix with three states for each of the five time-blocks. We are interested in the representative behavior of a worker but we have no way to peek into the representing meaning of each identified state, i.e., each latent state must be properly labeled.

Working on a pressing machine is different from making a purchase decision in shopping behavior. The activity cycle is only in the magnitude of seconds. We cannot ask the workers to fill a survey form while they are operating the pressing machine. Therefore, we must find a way to associate the HMM states to the human mind. We therefore took the transition matrix to its limiting steady distribution $\pi_i$ for each block $i$. Figure 5 reveals the resulting distributions of three states along the time progress of the five time-blocks. By doing so, the limiting distribution indeed showed a clear trend for the three states.

In terms of a prevalent heat map representation, the limiting probability values for each time block are color coded as the color bar listed on the right of the graph. The right panel of Figure 5 redraws the limiting distributions in a line graph. We can easily see a state remain high for all time and two other states switch probability in the middle of working day.

Observing the probability distributions in both panels of Figure 5, the state "Attention" is considered an early state in the cognition process and it was found to exhibit deep involvement in the conceptualization framework, as indicated in [6,7]. "Perception" is the one that many studies concentrate on. Song and Nakayama [8] argued that dynamic interaction exists between perception and decision [8]. The learning process must undergo a perception state before going to decision state [7,9]. The difference between the two states represent the incongruity that drives the learning process [9]. Therefore, we hypothesize that the labeling of the three states should be 1 = "Attention", 2 = "Decision", and 3 = "Perception". We will later verify this hypothesis through the feedback of factory production line.

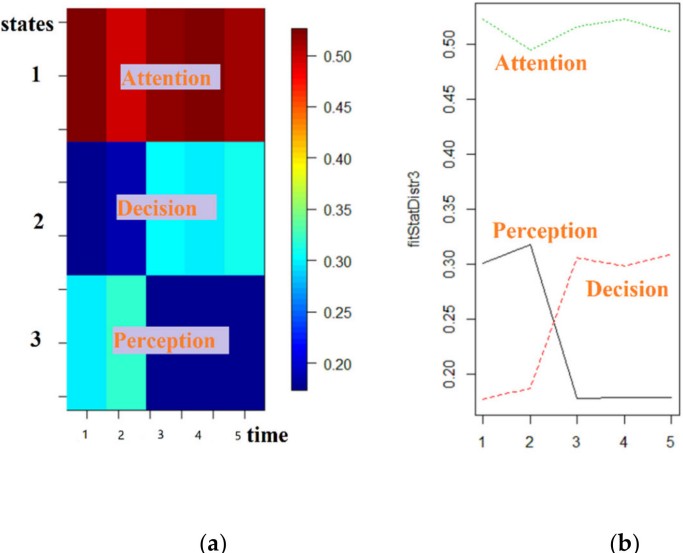

**Figure 5.** (**a**): Limiting steady distribution (color bar) of three cognitive states (vertical axis) over five time blocks (horizontal axis). (**b**): Probability changes of the three states over the five time blocks.

After labeling the three state in Figures 4 and 5, we observed the graph on the right panel of Figure 5. The cognitive mental states therefore emerge at the crossing point for the probability between "Perception" and "Decision." We found that even though the state of "Attention" remains over the entire period, "Perception" may not consistently follow. The operator initially remained perceptive most of time, but becomes reflexive afterward. The probability of "Perception" drops significantly in the middle of the day. Therefore, we surmised that a long-working operator became mentally fatigued and made decisions with less thinking.

## 4. Interacting with the Human Mind

We aim to ensure a modern working environment that is safe and reliable. The first goal ensures that mishaps does not occur in the process, while the second guarantees that a system can perform the required tasks for a period under a specific environment. We attempt to achieve these goals (in terms of human-machine interaction) by means of knowing the worker's mental state. In this section, we will describe how to interact with the human mind by first introducing the workflow in Petri Net (PN), followed by our proposed surrogate-assisted workflow.

### 4.1. Workflow in PN

PN is the graphical representation, modeling state transition, and dynamical relations among designated events. This tool has been widely used in the modeling of sophisticated industrial processes involving synchronous, asynchronous, and concurrent activities. Therefore, PNs are frequently used to analyze failure, deadlock, and reliability of dynamical processes. A PN is often represented in a tuple $G = (P, T, F, W, M_0)$, where $P$ is a finite set of places; $T$ is a finite set of transition; $F \rightarrow (P \times T) \cup (T \times P)$ is a set of arcs; $W : F \rightarrow \mathrm{N}$ represents the weight of the flow relation $F$; $M_0 : P \rightarrow \mathrm{N}$ is the initial marking vector, which represents the initial state of the system, and $P \cap T = \varnothing$ and $P \cup T \neq \varnothing$. An arc $f \in F$ connect places and transitions. We use a place to denote a state, while a transition denotes an operation for a modeled workflow. A $|P|$ number of tokens in each place exist under a state. Tokens flow through places, and the distribution of tokens is called a marking. A marking of $G$ is called a vector $m$. A PN with initial marking $m_0$ is denoted by $G(m_0)$. $t$ denotes the set of input places of transition $T$. A snapshot of tokens is central to the dynamics of a PN. A marking $m_2$ is reachable from $m_1$ if a firing sequence $s$ exists, thus bringing $m_1$ to $m_2$. The input tokens enable a transition based on the firing rule, which requires all input tokens to arrive simultaneously. A transition $t$ is enabled and

can be fired under $m$ iff $m(p) \geq F(p,t) \forall p \in P, t \in T$. After firing the transition, the tokens are consumed, but the transition will produce new tokens to all output places. A PN $G$ is called live if it is possible to fire any transition of $G$ starting from any marking $m_0$.

Figure 6 depicts an ordinary pressing workflow without involving a worker's internal mentality. Evidently, the workflow PN is live and suitable for modeling our work environment. The operator starts the repetitive work by inspecting a working piece. If the piece fails to surpass the finish quality standard (transition "DoWork"), then a worker will deliberate a pressing location and force to the working piece until the shape is sufficiently accurate. At present, a foot switch is in place to protect the operator from loss of attention. When only the foot switch is held, the transition "ReadyIn" can be fired. Transition "T5" will not be fired without going through an "FSoff" transition because of the toggle structure of PN. This mechanism prevents a single state to be consequently entered. The status will retain the toggle between two states repetitively.

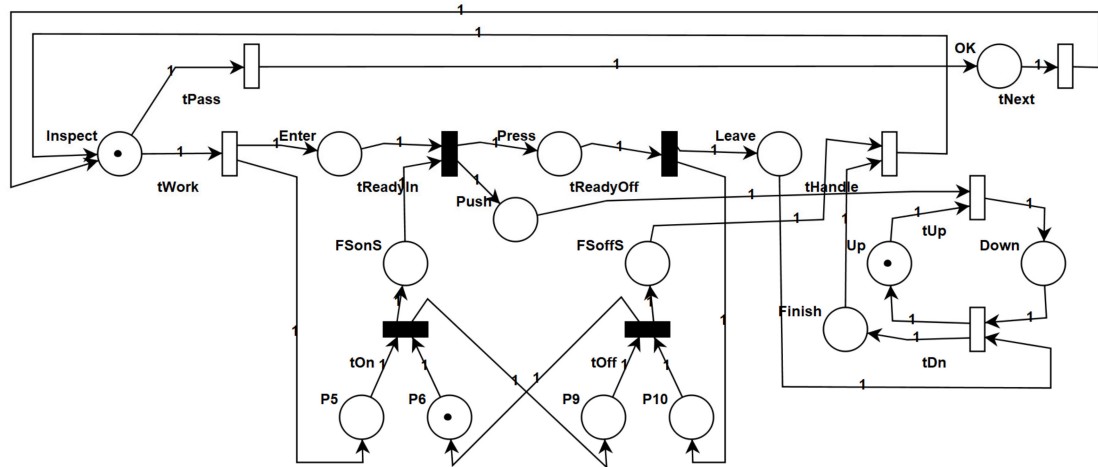

**Figure 6.** PN represents a pressing workflow for a mixed human–machine production line.

The press molding will not descend until three states are reached, namely, the workpiece is ready, the foot switch is on, and molding is currently in the up position. After completing the pressing, the piece goes to the "leave" state. The transition "Handle" will not be fired until the foot switch is turned off. After the transition, the workpiece is inspected and reworked if it fires the "DoPass" transition and undergoes the "OK" state.

### 4.2. Proposed Surrogate-assisted Workflow

An ordinary workflow may have insufficiencies in design. The foot switch remains under the same control of one's consciousness. If a fatigued worker does not concentrate on the operation of the hands, then a worker may be unable to ideally synchronize foot activity. Operating incidents or workpiece damage may occur unexpectedly.

To simultaneously address safety and productivity issues, we propose an augmented network by regulating the previous PN with identified states (see Figure 7). The pressing workflow is now synchronized by the transition of cognitive states. The "ReadyIn" transition will not be fired until the "Perception" state is identified. That is, operators must ensure its actions after deliberating the consequence. This check prevents a situation where an unconscious decision is made.

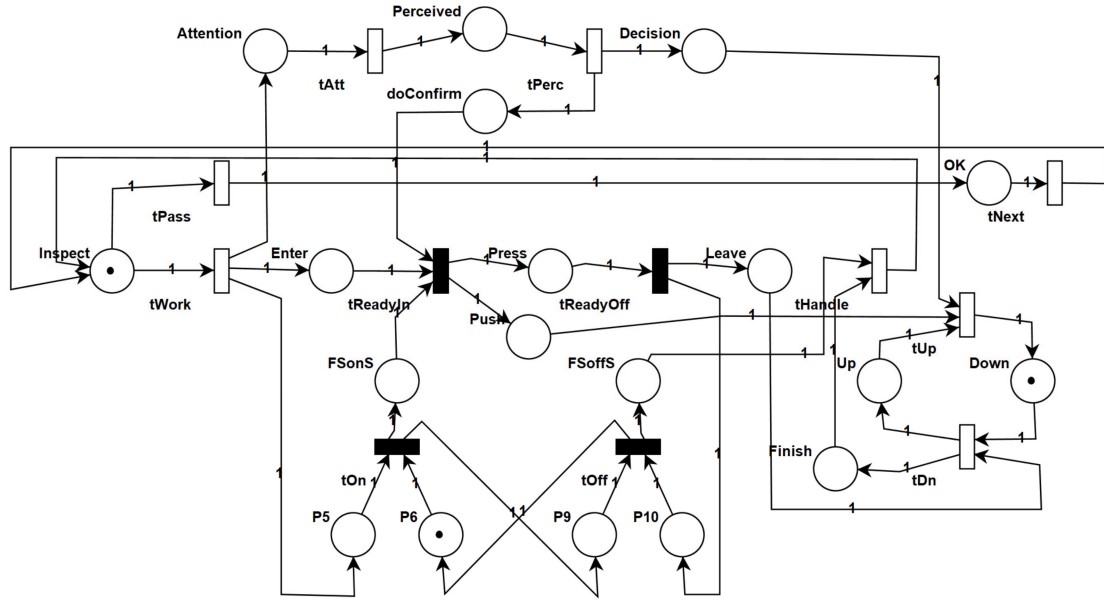

**Figure 7.** Surrogate-assisted PN interacting with the human mind.

In order to verify our hypothesis in this study, we propose to reinforce the PN workflow by integrating the cognitive states with affordable equipment. Figure 8 shows adoption of a wearable device attached to a worker's hat. By doing so, we were able to closely monitor the workflow, particularly for safety. For instance, if the worker turns his head away from the expected position but with his hands on the workpiece, it will trigger the system to not allow his foot activity to proceed. This is how our proposed method automatically interacts with the worker's state of mind to prevent injury from happening.

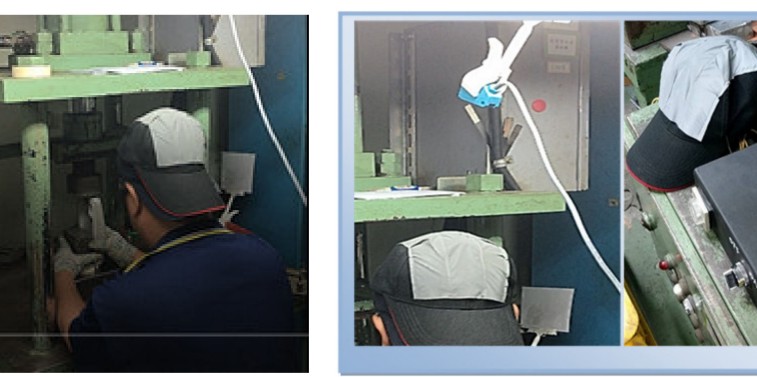

**Figure 8.** A worker was closely monitored by our proposed workflow with a wearable device.

To verify the effectiveness of our proposed method and the legitimacy of mind state labeling, following the validation procedure suggested in [24], we sent an evaluation form to the shop floor managers. The survey contained only questions: How do you rate the safety improvement after implementation of the equipment in Figure 8? How do you rate the productivity improvement after implementation of the equipment in Figure 8?

A total of 111 valid replies were collected; they have been summarized in Table 1. Among the 111 responders, 55% and 45% were male and female, respectively. Most responders were of 30-49 years of age and had 5-14 years of work experience. Almost all persons were first line workers and only 9 of them were engineers or managers. The numbers in Table 1 summarize the effectiveness of our device. In terms of expected value, the effectiveness was found to have 24% and 19% increase in safety and productivity, respectively. The standard deviations were 0.15 and 0.14, respectively. Therefore,

through feedback, we believe that the proposed PN in Figure 7 can maintain a balance of optimizing automation throughput and preventing safety threats when humans cooperate with machines.

**Table 1.** Summary of the effectiveness of our device.

| Survey Items | 0% | 10% | 20% | 30% | 40% |
|---|---|---|---|---|---|
| Safety improvement after the implementation | 21 | 10 | 21 | 17 | 42 |
| Productivity improvement after the implementation | 27 | 18 | 33 | 10 | 23 |

## 5. Conclusions

This study used a cognition-adaptive approach for the administrative control of human-machine safety interaction through IoT data. At first, we facilitated a Hidden Markov model (HMM) on top of a human cognitive model to capture the sequential faults of a production line worker suffering from work stress.

HMM was trained on the basis of the human cognitive model to determine the sequential faults of a production line worker. Accordingly, the trained model enabled us to surmise a person's mental state at production time. Although safety performance could not be evaluated in a limited period of time, production efficiency was found to be improved.

The most challenging part of this study was to find a way to estimate the human mental state. Hence, our main contribution was to use a surrogate-assisted PN to enhance human-machine safety interaction through real-time machine recording data. We found that safety and productivity increased. It is worthwhile to note that the PN automatically creates a safety alert whenever a fatigued worker stops following the proper operating procedure.

**Author Contributions:** Conceptualization, Y.J.C. and C.-N.C.; Methodology, Y.J.C.; Software, Y.J.C.; Validation, Y.J.C. and T.-K.L.; Formal Analysis, Y.J.C.; Investigation, Y.J.C. and T.-K.L.; Resources, C.-N.C. and T.-K.L.; Data Curation, T.-K.L.; Writing-Original Draft Preparation, Y.J.C.; Writing-Review & Editing, Y.J.C. and C.-N.C.; Visualization, Y.J.C.; Supervision, Y.J.C.; Project Administration, Y.J.C. and T.-K.L.; Funding Acquisition, C.-N.C. and T.-K.L.

**Funding:** This research was funded by Ministry of Science and Technology in Taiwan grant number 107-2221-E-992-092 and 107-2410-H-992-014-MY2.

**Conflicts of Interest:** The authors declare no conflicts of interest.

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
