# Peer review of "Human-Machine Interaction: Adapted Safety Assistance in Mentality Using Hidden Markov Chain and Petri Net"

_applsci, doi:10.3390/app9235066_

Round 1
Reviewer 1 Report
(1) This was a generally a well-written article about modeling human-machine interaction by Petri Nets (PN) and Hidden Markov Models (HMM). I am familiar with these modeling tools, but can claim no particular expertise on them; therefore, I cannot evaluate the accuracy or appropriateness of the models put forth in this paper.
(2) I can, however, critique other aspects of the paper. The first problem that I see is that the authors do not clearly describe how they may have accessed the perceptual, attentional, and decision-making processes of their human participants. I understand that their data came from IoT records, but the data only consist of observable events or actions, not the internal processes precipitating them. Therefore, there is no way to evaluate the validity of the models described.
(3) The Method-section is wholly inadequate to allow for evaluating the validity of this research. The figures 3 and 4 are very simple; in particular, Figure 4 cannot be interpreted as it lacks a key to the color coding as well as a description what the distributions of te cognitive states mean and how they were obtained.
(4) Similarly, the Results-section is quite unclear. Figure 9 is unreadable (too small) and its description does not provide enough information for interpretation. Ditto for Figure 10.
(5) The authors make several quite bold statements about their work: "...the trained model will enable us to estimate a person’s mental state at production time. Given the estimated cognitive states, we find that being attentive is easier than remaining perceptive for a production line worker." Even after several readings of this paper I cannot understand how the truthfulness of these claims can be assessed. Further, the authors state that "evaluation of shop floor workers reveals that our surrogate-assisted PN can enhance operation safety and improve production efficiency in the human–machine interaction setting." I saw no data reported about the results of "evaluation of shop floor workers" to afford such a conclusion. Perhaps the conclusions are based solely on the model results, but then the validity of the models is called to question.
(6) The paper could possibly be improved by clarifying the points I raised above. Moreover, the authors need to do a better job making a positive case for their work: Just what is their contribution to out understanding of human-machine interaction and of the tools to models it? As the paper is now, I am afraid I cannot recommend its publication.
Author Response
Response to Comments of applsci-591933
Paper Title: Human–machine interaction: Adapted safety assistance in mentality using hidden Markov chain and Petri net
Thank you very much for your thorough and constructive comments. We appreciate your efforts and time in reviewing our paper. We have worked hard to improve the contents and presentation of the paper. In the following document, we first described the key improvement of this revision, followed by the detailed responses to the referee's comments. For ease of your reference, your original comments were presented in italics followed by our response. The revised paragraphs in the main text were marked in blue ink. We hope that we have addressed all your comments and the paper is now publishable in Applied Sciences.
Key improvements in this revision
The abstract was revised to reflect our research purpose and findings. The section of introduction has been revised and reorganized carefully. The section of model development has been improved significantly. The section of research results is revised. The section of conclusion has been restated carefully to confirm contributions, research limit, and managerial implications.
Response to Reviewer 1's comments
This was a generally a well-written article about modeling human-machine interaction by Petri Nets (PN) and Hidden Markov Models (HMM). I am familiar with these modeling tools, but can claim no particular expertise on them; therefore, I cannot evaluate the accuracy or appropriateness of the models put forth in this paper.
Response:
We will do our best to respond to all of your comments.
I can, however, critique other aspects of the paper. The first problem that I see is that the authors do not clearly describe how they may have accessed the perceptual, attentional, and decision-making processes of their human participants. I understand that their data came from IoT records, but the data only consist of observable events or actions, not the internal processes precipitating them. Therefore, there is no way to evaluate the validity of the models described.
Response:
In response to your comment, We have added more description about the process of obtaining cognitive states in section 3.3 on page 5. We understand the difficulty to collect the internal mind states in a fashion of real-time and working environment. Hidden Markov chain can be trained with discretized output symbols to get the transition matrix but cannot get the interpretation of each state. In this revision, we explained the steps of obtaining the labeling of internal states. On page 6, we observe our results by matching the limiting distributions to the evidence of existing studies. Our limiting probabilities are found consistent with existing literature resulting in a reasonable estimate in the state labeling.
The Method-section is wholly inadequate to allow for evaluating the validity of this research. The figures 3 and 4 are very simple; in particular, Figure 4 cannot be interpreted as it lacks a key to the color coding as well as a description what the distributions of the cognitive states mean and how they were obtained.
Response:
We have already made proper changes in section 3. A more detailed exposition has been added. Regarding Fig 3 and Fig. 4, we enhanced the description to these two figures. The two graphs in the manuscript that describe the limiting distribution have been merged into Fig. 5 for better readability.
Similarly, the Results-section is quite unclear. Figure 9 is unreadable (too small) and its description does not provide enough information for interpretation. Ditto for Figure 10.
Response:
We decide to delete Figs. 9 and 10 because the information provided was irrelevant to the rest of exposition. We think that the reachability study can be future work by itself.
The authors make several quite bold statements about their work: "...the trained model will enable us to estimate a person’s mental state at production time. Given the estimated cognitive states, we find that being attentive is easier than remaining perceptive for a production line worker." Even after several readings of this paper I cannot understand how the truthfulness of these claims can be assessed. Further, the authors state that "evaluation of shop floor workers reveals that our surrogate-assisted PN can enhance operation safety and improve production efficiency in the human–machine interaction setting." I saw no data reported about the results of "evaluation of shop floor workers" to afford such a conclusion. Perhaps the conclusions are based solely on the model results, but then the validity of the models is called to question.
Response:
Since we have no effective means to qualify human’s state of mind, we think that a survey to the shop floor workers will help speak for the effectiveness of our proposed model. Accordingly, we have revised our conclusions.
The paper could possibly be improved by clarifying the points I raised above. Moreover, the authors need to do a better job making a positive case for their work: Just what is their contribution to out understanding of human-machine interaction and of the tools to models it? As the paper is now, I am afraid I cannot recommend its publication.
Response:
We hope that we have explained clearly on the process for human-machine interaction. Without directly measuring human brain’s thinking, we think that increased productivity can indirectly validate the effectiveness of our model.
Response to Reviewer 2's comments
The study focuses on the development of a hidden Markov model combined with a human cognition model to improve the operational safety of a pressing process. Overall the manuscript is well written, based on a novel concept and is industrially relevant.
Response:
We will do our best to respond to your comments.
Replace the word 'process safety' with 'personnel safety' in the line above figure 2. Process safety deals with reducing the risk associated with a chemical process from a societal perspective. The difference between process safety and personal safety is that process safety tends to focus on mitigating risks through the inherent design of a system, whereas personal safety focuses on enforcing behavioral changes in individual workers and teams in order to prevent incidents.
Response:
We have changed the wording in the last line of page 3.
Increase the size of Figure 2 to improve legibility.
Response:
We have increased the size of Fig. 2.
Response to Reviewer 3's comments
This paper addresses the interesting topic of how to model human cognitive states to improve human-machine coordination in industry. The use of Hidden Markov Chain and Petri nets models sounds relevant and integrates a set of previous existing researches.
Response:
We will do our best to respond to your comments.
Firstly, the abstract and the introduction must be more structured in order to provide a clear presentation of the context and the challenges addressed by this study.
Response:
The abstract has been revised to be better structured.
Some formulations are unclear, like 'machine feedback big data'.
Response:
We have changed the wording to IoT data on page 2.
A more structured abstract including the issue, current works, objective/proposal, method, results must be written.
Response:
. This comment is very similar to your second comment. We have significantly revised the abstract to reflect your concern.
In the introduction, relations between some statements are difficult to follow, e.g.’Hence, administrative means are common in the old era to regulate the habit of human workers and prevent work hazards’.
Response:
We have rewritten the sentence on page 1.
The term ‘However’ is used third times in the second paragraph of the introduction...
Response:
We have rephrased the sentence.
The reader shows difficulties to represent how the authors reach their findings about the weights given to each hidden cognitive state (perception, decision, attention). How do you distinguish the cognitive states in the processing produced by your model? What are the relations between concrete behavioral cues and these hidden states?
Response:
We have added more description about the process of obtaining cognitive states in section 3.3 on pages 5 and 6.
Reviewer 2 Report
The study focuses on the development of a hidden Markov model combined with a human cognition model to improve the operational safety of a pressing process. Overall the manuscript is well written, based on a novel concept and is industrially relevant.
The following minor changes will further improve the quality of the manuscript:
Replace the word 'process safety' with 'personnel safety' in the line above figure 2. Process safety deals with reducing the risk associated with a chemical process from a societal perspective. The difference between process safety and personal safety is that process safety tends to focus on mitigating risks through the inherent design of a system, whereas personal safety focuses on enforcing behavioral changes in individual workers and teams in order to prevent incidents.
Increase the size of Figure 2 to improve legibility.
Author Response

(The authors gave the same response as above.)

Reviewer 3 Report
This paper addresses the interesting topic of how to model human cognitive states to improve human-machine coordination in industry.
The use of Hidden Markov Chain and Petri nets models sounds relevant and integrates a set of previous existing researches.
Two main weakness have been detected in the manuscript.
1/Firstly, the abstract and the introduction must be more structured in order to provide a clear presentation of the context and the challenges addressed by this study.
Some formulations are unclear, like 'machine feedback big data'.
A more structured abstract including the issue, current works, objective/proposal, method, results must be written.
In the introduction, relations between some statements are difficult to follow, e.g.’Hence, administrative means are common in the old era to regulate the habit of human workers and prevent work hazards’.
The term ‘However’ is used third times in the second paragraph of the introduction...
2/ The reader shows difficulties to represent how the authors reach their findings about the weights given to each hidden cognitive state (perception, decision,attention). How do you distinguish the cognitive states in the processing produced by your model? What are the relations between concrete behavioral cues and these hidden states?
Author Response

(The authors gave the same response as above.)
